# Enzymatic Hydrolysis of *Tenebrio molitor* (Mealworm) Using Nuruk Extract Concentrate and an Evaluation of Its Nutritional, Functional, and Sensory Properties

**DOI:** 10.3390/foods12112188

**Published:** 2023-05-29

**Authors:** Legesse Shiferaw Chewaka, Chan Soon Park, Youn-Soo Cha, Kebede Taye Desta, Bo-Ram Park

**Affiliations:** 1Department of Agro-Food Resources, National Institute of Agricultural Science, Rural Development Administration, Wanju 55365, Republic of Korea; legeshiferaw@gmail.com (L.S.C.); hipcs@korea.kr (C.S.P.); 2Department of Food Science and Nutrition, Jeonbuk National University, Jeonju 54896, Republic of Korea; cha8@jbnu.ac.kr; 3National Agrobiodiversity Center, National Institute of Agricultural Science, Rural Development Administration, Jeonju 54874, Republic of Korea; kebedetdesta@korea.kr

**Keywords:** protein hydrolysate, *Tenebrio molitor*, nuruk, protease, alcalase, flavourzyme

## Abstract

Enzymatic protein hydrolysis is a well-established method for improving the quality of dietary proteins, including edible insects. Finding effective enzymes from natural sources is becoming increasingly important. This study used nuruk extract concentrate (NEC), an enzyme-rich fermentation starter, to produce protein hydrolysate from defatted *Tenebrio molitor* (also called mealworm, MW). The nutritional, functional, and sensorial properties of the hydrolysate were then compared to those obtained using commercial proteases (alcalase and flavourzyme). The protease activities of the crude nuruk extract (CNE), NEC, alcalase, and flavourzyme were 6.78, 12.71, 11.07, and 12.45 units/mL, respectively. The degree of hydrolysis and yield of MW hydrolysis by NEC were 15.10 and 35.92% (*w*/*w*), respectively. MW hydrolysate was obtained using NEC and had a significantly higher free amino acid content (90.37 mg/g) than alcalase (53.01 mg/g) and flavourzyme (79.64 mg/g) hydrolysates. Furthermore, the NEC hydrolysis of MW increased the antioxidant and angiotensin-converting enzyme inhibitory activity, with IC_50_ values of 3.07 and 0.15 mg/mL, respectively. The enzymatic hydrolysis also improved sensory properties, including umaminess, sweetness, and saltiness. Overall, this study found that the NEC hydrolysis of MW outperformed commercial proteases regarding nutritional quality, sensory attributes, and biological activity. Therefore, nuruk could potentially replace commercial proteases, lowering the cost of enzymatic protein hydrolysis.

## 1. Introduction

In reaction to a future protein shortage, insect-based proteins are being viewed as a viable solution for feeding a projected 9.7 billion people by 2050 [1]. To this end, edible insects are recommended as alternative protein sources because they efficiently convert nutrients, require less land for production, and are more eco-friendly than animal proteins [2,3]. *Tenebrio molitor*, commonly known as mealworm (MW), is an insect that is mainly used as food and feed in its larval stage. It is the most widely bred and traded edible insect, with a high feed conversion ratio of 3.4 to 6.1 kg of feed ingested per kg of harvested larvae [4]. It can compete with animal-based protein sources due to its high protein content (~50%) and balanced amino acid profiles [5].

Insects are believed to be more acceptable to consumers when they are fragmented and used as an ingredient in foods rather than in their whole form [3,6,7]. However, since merely pulverized dry MW powder is insoluble in water, its application as a protein supplement material would be very limited. Therefore, protein hydrolysis techniques are critical for the use of insects as an alternative protein source and for ensuring their palatability [8].

Edible insects have a rigid outer wall due to the development of structural proteins such as chitin, which contribute to the rigidity of the insect cuticle and serve as an attachment matrix for other cuticle proteins [9]. On the other hand, enzymatic protein hydrolysis is a well-established processing method that can improve the nutritional, functional, and sensory properties of food proteins [2,10,11,12,13,14,15]. Several studies have shown that proteolytic enzyme hydrolysis produces a hydrolysate with a superior nutraceutical profile compared to acid/alkaline hydrolysis [10,12]. Enzymes can increase the contents of proteins extracted in the forms of peptides and free amino acids by enhancing their solubility, decreasing the molecular mass, and increasing repulsive interactions between peptides and hydrogen bonds [16]. As a result, such an effect can improve their absorption rate in the small intestine and reduce allergenic effects [17].

Several reports on the enzymatic hydrolysis of edible insect proteins using commercial proteases are available in the literature. Leni et al. [16] reported that the protein extraction yield, the degree of hydrolysis (DH), and the released free amino acids (FAA) of MW could be improved using enzyme hydrolysis. The improvement of the amino acid profile and scavenging potential of MW using alcalase and flavourzyme hydrolysis were also investigated [18]. These commercial enzymes could also be applied to improve the techno-functional properties of the migratory locust (*Locusta migratoria* L.) protein flour [11]. However, the cost and availability of commercial enzymes are limiting factors for their complete application in the hydrolysis of edible insect proteins. Therefore, there is a growing interest in exploring and evaluating potential enzyme sources [17].

Nuruk, a traditional fermentation starter agent, is rich in hydrolytic enzymes produced by molds, which convert starch and protein into fermented sugar and amino acids [19,20,21]. Through the mold’s growth and development in nuruk, various enzymes, such as protease, amylase, and lipase, are produced at the terminus of the mycelium. It is widely used for manufacturing various Asian-fermented alcoholic beverages [22,23]. Although previous studies have shown that nuruk has high protease activity, its applications for protein hydrolysis are limited [20,24].

As a result, the purpose of this work was to investigate the capacity of nuruk for the enzymatic hydrolysis of MW protein in comparison to two popular commercial enzymes (alcalase and flavourzyme), and to characterize the nutritional, sensory, and functional aspects of MW hydrolysate.

## 2. Materials and Methods

### 2.1. Materials, Chemicals, and Reagents

Dried MW (*T. molitor*) larvae were obtained from the National Academy of Agricultural Sciences Insectarium (Jeonju, Jeolabuk-do, Republic of Korea) and defatted using a pressing method. All the reagents and chemicals used were of analytical grade and were commercially available.

Kereyang nuruk (Bio Nurook, Hwasung, Republic of Korea) and commercial protease enzymes (alcalase 2.4 L FG and flavourzyme 1000 L) were obtained from Novozyme (Bagsvaerd, Denmark).

Laboratory reagents and kits, including the AccQ·Tag ultra derivatization kit and amino acid standards, were purchased from Waters (Milford, MA, USA), whereas casein and 2,4,6-trinitrobenzene sulfonic acid (TNBS) were obtained from Sigma Aldrich (St. Louis, MO, USA). The angiotensin-converting enzyme (ACE) assay kit was obtained from Abcam (Cambridge, UK), while an Amicon ultra centrifugal filter 30 kDa was purchased from Merck (Darmstadt, Germany).

### 2.2. Preparation of Nuruk Extract Concentrate and Protease Activity

Nuruk’s enzymes were extracted using a previously reported method with slight modification [25]. Briefly, nuruk was extracted for 1 h at 37 °C with agitation at 80 rpm using deionized water in a 1:10 (*w*/*v*) ratio. The homogenate was centrifuged at 5000 rpm for 15 min at 4 °C [25]. The supernatant (crude nuruk extract) was then transferred to centrifugal filters (30 kDa), where it was centrifuged at 7119× *g* for 30 min at 4 °C. The filtrate was discarded, and the retentate was a nuruk extract concentrate. The NEC was stored at −80 °C if not used.

Proteolytic activities of the crude nuruk extract, nuruk extract concentrate (NEC), alcalase, and flavourzyme were separately assayed using TNBS as previously reported with some modifications [26]. Briefly, 5 mL of a 0.5% casein solution in a 0.21 M sodium phosphate buffer (pH 8.20) was incubated with the addition of 1 mL of the extracted enzyme sample for 30 min at 37 °C with agitation at 80 rpm. Next, 2 mL of the hydrolysate was mixed with 18 mL of pre-warmed sodium dodecyl sulfate (SDS, 1%) and incubated at 75 °C for 20 min with agitation at 80 rpm. This was followed by the addition of the hydrolysate and SDS mixture (0.25 mL) into a solution containing 2 mL of a 0.21 M sodium phosphate buffer (pH 8.20) and 2 mL of 0.1% TNBS. The sample was then incubated for another 1 h at 50 °C with agitation at 80 rpm. A total of 4 mL of 0.1 N HCl was then added to stop the reaction, and the sample was allowed to cool for 30 min.

Absorbance was measured at 340 nm using a Multiskan Sky spectrophotometer (Thermo Fisher Scientific Vantaa, Finland). The standard calibration curve was made from serial dilutions of the leucine solution (0.25, 0.5, 1, and 2 µmol/mL), and 1 unit of enzyme activity was defined as the micromoles of leucine liberated per ml of nuruk or protease enzyme in 30 min.

### 2.3. Enzymatic Hydrolysis of MW and Hydrolytic Properties of NEC

The enzymatic hydrolysis of MW was carried out using a previously published methodology with some modifications [26]. Initially, 20 g of the defatted MW powder (71.41% protein) was added to a 500 mL flask containing 150 mL of deionized water. Next, the mixture was allowed to hydrate for 10 min at room temperature with gentle mixing. NEC (10%, 2 mL), alcalase (1%, 200 µL), and flavourzyme (1%, 200 µL) were added to the MW protein solution independently (Table 1).

In a separate reaction flask, the solution was agitated in a shaker incubator for 6 h at 80 rpm, 50 °C, and pH 7. The hydrolysate solution was then heated for 15 min at 85 °C to inactivate the enzyme, which was then centrifuged at 8000 rpm for 20 min at 4 °C to separate the supernatant before being lyophilized as protein hydrolysate [26]. Following that, the enzymatic hydrolysis features were characterized by the degree of hydrolysis, total soluble solids, hydrolysis yield, and molecular size distribution, as described below.

#### 2.3.1. Degree of Hydrolysis

The degree of hydrolysis (DH), which can be defined as the percentage of cleaved peptide bonds from the total peptide bonds, was calculated using the TNBS method [26]. An aliquot of hydrolysate samples (100 μL) at the beginning (0 h) and at the end of hydrolysis (6 h) was added to 900 μL of 5% (*w*/*v*) SDS. Following that, it was heated for 5 min at 85 °C to inactivate the enzyme and measure the degree of hydrolysis. Duplicate aliquots (0.25 mL) of the test or standard solutions were added to the test tubes containing 2.0 mL of a sodium phosphate buffer (0.2 M, pH 8.2). In total, 2 mL of the TNBS reagent (0.1% (*w*/*v*) was then added to each tube, followed by mixing and incubation at 50 °C for 60 min in a covered water bath. After incubation, the reaction was stopped by adding 4.0 mL of 0.1 N HCl.

The samples were then allowed to cool at room temperature for 30 min before absorbance values were measured at 340 nm using a Multiskan Sky spectrophotometer (Thermo Fisher Scientific Vantaa, Finland). L-Leucine (0.25, 0.5, 1, and 2 µmol/mL) was used to generate a standard curve, and the DH value was calculated using the following formula:(1)$$\mathrm{DH}=\frac{\left(\mathrm{AN}1-\mathrm{AN}2\right)}{N}\times 100$$
where AN1 and AN2 are the amino nitrogen content before and after hydrolysis (mg/g protein), respectively, and N is the nitrogen content of the peptide bonds in the MW protein (mg/g protein).

#### 2.3.2. Total Soluble Solid and Hydrolysis Yield

The total soluble solid content in the supernatant (hydrolysate) was measured at the end of enzymatic hydrolysis using a refractometer (HI96811, Hanna Instruments Inc., Nusfalau, Romania).

The hydrolysis yield was the weight of soluble protein extracts dried after hydrolysis, which was calculated using the following equation (Equation (2)).
(2)$$Yield\;\left(\%\right)=\frac{Weight\;in\;g\;of\;freez\;dried\;hydrolysate}{weight\;in\;g\;of\;sample}\times 100$$

#### 2.3.3. Molecular Weight Distribution

The molecular weight distribution of MW hydrolysates was obtained by Size Exclusion Chromatography (SEC) using High-Performance Liquid Chromatography equipped with a TSKgel G3000PWxl column (7.8 mm I.D × 30 cm, 7 µm) and a DAD detector (Agilent infinity 1260 II LC system, Santa Clara, CA, USA). Briefly, 10 mg/mL of each MW hydrolysate solution and 1 mg/mL of the standard (1–670 kDa) mixture were eluted using a phosphate buffer (pH 7.4) at a flow rate of 0.5 mL/min and were detected at 220 nm. Calibration curves were constructed using Throglobulin (670 kDa), γ-globulin (150 kDa, Ovalbumin (45 kDa), Myoglobin (17 kDa), Aprotinin (6.7 kDa), and Angiotensin II (1 kDa) as the protein standards. The peptide molecular mass distribution was estimated from a logarithmic calibration curve that was constructed by plotting the elution time (in min) against MW (in kDa) (Appendix A) and the peak area to estimate the proportion of each molecular weight of the peptides present.

### 2.4. Nutritional Composition of MW and Its Hydrolysates

#### 2.4.1. Proximate and Total Amino Acid Composition

The moisture, fat, total ash, and nitrogen contents of whole and defatted MW were determined using the Association of Official Analytical Chemists (AOAC) standard procedures [27]. In summary, oven drying at 105 °C was used to measure the moisture content, while the ash content was evaluated by incineration in a furnace at 550 °C. The lipid content was assessed using the ether extraction method, while the total nitrogen was determined by a Kjeldahl system and converted into protein using a factor of 6.25. The carbohydrate content was estimated using the difference method by subtracting the sum of crude protein, lipid, moisture, and ash contents from 100.

For the total amino acid analysis, 0.5 g of whole and defatted MW was suspended in 5 mL of 6 N HCl and heated at 110 °C overnight. Following that, it was centrifuged for 15 min at 4 °C at 2781× *g*. The supernatant was diluted 1/10, and its pH was adjusted to 7 before it was derivatized using an AccQ·Tag derivatization kit according to the manufacturer’s instructions [28] (Waters, Milford, MA, USA). Briefly, 70 μL of the preheated AccQ·Tag borate buffer was mixed with 20 μL of the derivatizing reagent. 10 μL of the sample was added, vortexed, and heated using a heating block for 10 min at 55 °C.

The total amino acid contents of the samples were determined using an Ultra Performance Liquid Chromatography (ACQUITY UPLC system, Waters Corporation, Milford, MA, USA) equipped with ACCQ-TAG ULTRA C18 column (Ø 2.1 × 100 mm, 1.7 µm) and photodiode array (PDA) detector. During the analysis, 10 µL of the derivatized samples and mixtures of amino acid standards were injected into the UPLC-PDA. The analysis time, flow rate, and detector wavelength were 10 min, 0.7 mL/min, and 260 nm, respectively. The quantification of individual amino acids was conducted using external calibration curves from 17 pure amino acid reference standards by Empower software (Waters, Milford, MA, USA).

#### 2.4.2. Free Amino Acid Analysis

For free amino acid content analysis, a 0.5 g sample was suspended in 5 mL of deionized water for 30 min, followed by centrifugation for 10 min (4 °C, 2781× *g* rpm). The supernatant was derivatized and analyzed using a similar method as the total amino acid analysis.

### 2.5. Biological Activities of MW Hydrolysate

#### 2.5.1. Antioxidant Activity

The antioxidant activity was measured using a 2,2′-azino-bis (3-ethylbenzothiazoline-6-sulphonic acid (ABTS) radical-scavenging assay as described before [29], with slight modifications. Briefly, equal volumes of a 7.4 mM ABTS solution and 2.6 mM potassium persulfate solution were mixed and kept in the dark for 14–16 h at room temperature to form ABTS radical stock solution. The mixture was then diluted with 10 mM phosphate-buffered saline (PBS, pH 7.4), and the absorbance at 734 nm was adjusted to 0.70 ± 0.02. Next, 10 µL of the sample was mixed with 190 µL of the ABTS radical solution and was allowed to react in the dark at room temperature for 10 min. The absorbance was measured at 734 nm. The percentage of ABTS radical-scavenging activity was determined according to the Equation (3) below, and the IC_50_ was determined from serial concentration samples (1, 2, 5, and 10 mg/mL).
(3)$$\mathrm{ABTS}\;\mathrm{inhibition}\;\left(\%\right)=\frac{\left(A\right)\mathrm{control}\;-\left(A\right)\mathrm{Sample}}{\left(A\right)\mathrm{control}}\times \;100$$

#### 2.5.2. Angiotensin Converting Enzyme Inhibitory Activity

The ACE inhibitory activity of hydrolysate was determined in vitro using the ACE assay kit (abcam, Cambridge, UK) according to the manufacturer’s instructions. In total, 20 µL of hydrolysate, 10 µL of ACE, and 50 µL of *O*-aminobenzoyl peptide (Abz-based peptide) were used as a substrate to release a fluorophore and were mixed in a black 96 well plate. Fluorescence (Ex/Em = 330/430) was measured in a kinetic mode for 2 h at 37 °C, and the slopes of all samples, including the control (without inhibitor/hydrolysate), were calculated by dividing the change in the relative fluorescence unit (ΔRFU = RFU_2_ − RFU_1_) with the time Δt (t_2_ − t_1_). The inhibitory activity (%) was then calculated as follows (4).
(4)$$\mathrm{ACE}\;\mathrm{Inhibition}\;\left(\%\right)=\frac{slope\;of\;C-Slope\;of\;S}{Slope\;of\;C}\;\times \;100$$
where the slope of C represents the difference in the relative fluorescence unit measured at a reaction time of ACE (enzyme) with the substrate, and the slope of S represents ACE and the substrate in the presence of MW hydrolysate. The IC_50_ of the MW hydrolysate was determined from a serial concentration of the samples (1, 0.5, and 0.25 mg/mL).

### 2.6. Sensory Properties of MW Hydrolysate Using E-Tongue

The sensory properties of the defatted MW and its hydrolysate (0.5% *w*/*v*) were measured using a potentiometric E-tongue (αAstree, Alpha M.O.S., Toulouse, France). ASTREE is an electronic tongue based on the potentiometric measurement principle using taste-sensing electrodes and is dedicated to taste analysis. The sensors are designed to mimic the human tongue, and they can differentiate between different tastes and flavors, such as sweet, sour, salty, bitter, and umami, based on changes in their electrical properties in response to different chemicals or molecules [30]. Before starting the analysis, conditioning, calibration, and diagnostic steps were performed using the previously reported protocol [31]. In summary, a solution of hydrochloric acid (0.01 mol/L) was used for conditioning and calibration, whereas hydrochloric acid (0.01 mol/L), sodium glutamate (0.01 mol/L), and sodium chloride (0.01 mol/L) were used for the diagnostic step [31]. The scale of all taste attributes could be defined between 0 and 10, where 0 referred to the least intense and 10 was the most intense taste perception.

### 2.7. Statistical Analysis

The results were presented as the mean ± standard deviation from triplicate measurements. An analysis of variance was conducted using XLSTAT software version 2019.2.2 (Lumivero, Denver, CO, USA), and differences at *p* ≤ 0.05 were considered significant. Taste scoring was conducted using Alpha software (Version 12.4., Alpha M.O.S., 2012).

## 3. Results and Discussion

### 3.1. Hydrolytic Properties of Nuruk Extract Concentrate

#### 3.1.1. Protease Activity

Due to nuruk being a complex microbial ecosystem that supports the growth and production of microorganisms and enzymes during natural fermentation [21,32,33], we conducted proteolytic activity measurements on both commercial enzymes and nuruk extracts, which were used to hydrolyze mealworms. Table 2 summarizes the protease activity results for each enzyme in enzyme unit per mL (EU/ mL) and enzyme unit per gram (EU/g). The protease activity of NEC was 127.08 EU/g which was approximately two-fold compared to the crude nuruk extract (67.84 EU/g). This increase in protease activity could be due to ultrafiltration using 30 kDa [34]. In addition, the crude extract contained not only enzymes but acidic substances and microbe metabolites that may deteriorate in hydrolysate taste. The observed nuruk’s protease activity (67.84 EU/g) was lower compared to a previous study that found nuruk made from wheat and soybean to have a maximum of 84.38 EU/g of dry-weight protease activity [24]. This might be due to the inclusion of soybean as an ingredient to prepare nuruk in the study [22].

Jang-Eun et al. [20] conducted an evaluation of protease activities in 46 distinct types of nuruk, revealing a range of 1.42–10.32 EU/mL. Interestingly, these findings align with the protease activity observed in our study on crude nuruk, which measured 6.78 EU/mL. NEC’s protease activity in EU/mL was greater than that of alcalase and flavourzyme. The variation in the former was significantly different (*p* ≤ 0.05), but there was no significant difference between NCE and flavourzyme. These two commercial enzymes cleave proteins in opposite ways, with flavourzyme being an endo and exopeptidase and alcalase being an endopeptidase [35]. Overall, the protease activity data clearly showed that 10% of NEC would have approximately equivalent activity with alcalase (1%) or flavourzyme (1%). Based on this, one could estimate the enzyme dose used in the enzymatic hydrolysis of MW.

#### 3.1.2. Degree of Hydrolysis (DH), Hydrolysate Yield, and Total Soluble Solid

The DH value was an indicator of the cleavage of peptide bonds and the breakdown of the complex and structured proteins into smaller peptides [11]. The DH of MW by NEC, alcalase, and flavourzyme for 6 h are presented in Table 3. The highest DH was achieved by NEC (15.10%), followed by flavourzyme (11.03%) and alcalase (9.56%). In contrast, the DH of the control was very low (0.55%), indicating a lack of the endogenous protease in defatted MW. This clearly shows that the DH recorded by NEC was due to the presence of protease enzymes that effectively broke down the protein chains [26].

Leni et al. [36] found a slightly higher DH of 14.90% with the alcalase hydrolysis of mealworms for 5 h, which could be due to the method used to determine DH (pH state method) while our method was TNBS [26]. Purschke et al. [11] reported a slightly higher DH of 17.10 and 16.40% on the hydrolysis of migratory locusts with 1% alcalase and 1% flavourzyme for 8 h, respectively, and this variation might be due to the different insect species [16].

The highest total soluble solid level was recorded for alcalase hydrolysis (7.97%), followed by nuruk (5.97%), flavourzyme (5.70%), and control (3.60%). This indicated an improvement in solubility due to enzymatic hydrolysis [11]. Similarly, the hydrolysis of MW using NEC yielded 35.92% powder, which was lower than alcalase hydrolysis (42.88%) but significantly higher than flavourzyme hydrolysis (28.99%). Moreover, NEC hydrolysis resulted in a two-fold yield compared to water extraction (control) (*p* ≤ 0.05).

In line with our study, Yu et al. [37] reported hydrolysis yields of 42.05 and 26.27% using alcalase and flavourzyme, respectively. Similarly, a 38.70% hydrolysis yield from MW was reported following alcalase treatment for 8 h [18]. The high hydrolysis yield and total soluble solid level, as well as the low DH, could be attributed to the fact that alcalase is an endopeptidase, which aids in the cleavage of internal peptide bonds in proteins, resulting in small chains with fewer amino groups [35].

#### 3.1.3. Molecular Weight Distribution

The MW hydrolysate pattern and the resultant protein peptide size are shown by SEC analysis (Figure 1a). The SEC chromatogram indicated that all enzymes treated MW hydrolysate exhibited a wide range of peptide sizes distribution from <1 kDa to >670 kDa and various peak elution patterns. The defatted MW indicated a very small amount of the native peptide MW protein. Additionally, the MW hydrolysis control sample (hydrolysis without enzyme) chromatogram, two peaks higher than 670 kDa and with a lower peak than other hydrolysate samples smaller than 6.7 kDa, were present. This suggested that the enzymatic hydrolysis was effective in generating small molecular weight peptides (6.7–1 kDa and ≤1 kDa). In particular, MW-A was found to have the highest content of peptides with a size of 1 kDa or less. This observation can be attributed to the higher generation of soluble peptides resulting from the endo-protease activity of alcalase [13]. This was consistent with the findings of Yu et al. (2017), who reported that alcalase effectively generates small-size peptides [37].

We categorized and compared all the hydrolysate peptide sizes based on the elution time of the protein standards (670 kDa, 150 kDa, 45 kDa, 17 kDa, 6.7 kDa, 1.7 kDa, and 1 kDa), as shown in Figure 1b. The area of the chromatograms was plotted as a bar graph by dividing the peptide size into 1 kDa or less, 1–6.7 kDa, and 670 kDa or more and summing up the peak area. MW-A contained the highest concentration of 1–6.7 kDa peptides, followed by MW-NEC and MW-F, respectively. Several studies have reported isolation of peptides by size to identify the peptide molecular weight and to evaluate their biological activities [37,38,39,40,41]. All the hydrolysate, characterized by a high content of peptides in the size range of 1–6.7 kDa and less than 1 kDa, was highly correlated with their biological activities in Section 3.3 (see also Appendix A).

### 3.2. Nutritional Composition

#### 3.2.1. Proximate and Total Amino Acid Composition of MW

The proximate and total amino acid contents of the whole and defatted MW are summarized in Table 4. In the defatted sample, the crude protein content was higher (71.41%) compared to the whole sample (49.16%). These values fall within the range reported in the literature for mealworm proteins, which varies from 44% to 52% [3,5,7,42,43]. Therefore, the findings suggest that mealworms have the potential to serve as a valuable alternative protein source.

Interestingly, the defatted mealworms exhibited higher total and free amino acid contents. However, it is noteworthy that both MW samples exhibited low free amino acid compositions, likely due to the limited solubility of MW protein in deionized water at pH 7 Appendix A [18]. In contrast, the carbohydrate and crude fat contents were higher in the whole mealworm sample. The total amino acid contents whole and defatted MW were 39.62% and 53.48%, respectively. This difference between protein content and total amino acid content may be attributed to the presence of non-protein nitrogen compounds, such as chitin, nucleic acids, phospholipids, and excretion products [8].

The total amino acid compositions of both the whole and defatted mealworm (MW) samples are provided in Appendix A. In line with the guidelines established by the World Health Organization (WHO) for protein foods [16], all essential amino acids in the MW samples meet these requirements, as indicated in Appendix A. In a study by Azagoh et al. [44], a slightly lower amino acid content was reported for *Tenbrio molitor* (mealworm), which was compared with other protein-rich food sources. Our study findings are consistent with previous studies and underscore the potential of MW as an alternative protein source [18,44,45]. These observations highlight the nutritional composition of mealworms and their potential as a sustainable protein source. The higher protein content and favorable amino acid profile, especially in the defatted mealworms, further emphasizes their suitability for various food applications.

#### 3.2.2. Free Amino Acid Composition of MW Hydrolysate

The free amino acid concentrations of MW hydrolysate are provided in Table 5. All enzymatic hydrolysis improved the free amino acid composition of MW compared to the defatted MW. It was reported that enzymatic hydrolysis improved protein solubility and generated free amino acids and low molecular weight peptides, which enhanced its bioavailability [12]. Several researchers have reported free amino acid generation through the enzymatic hydrolysis of edible insects, including MW [11,12,13,16,18]. Among the essential amino acids, valine, leucine, and lysine were the most abundant. By contrast, methionine was the least abundant amino acid in all the hydrolysates. Following the enzyme difference used in the hydrolysate preparation, significant differences in the levels of both essential and non-essential amino acids were observed (*p* ≤ 0.05).

With a few exceptions, NEC-treated MW hydrolysate outweighed the two commercial enzymes in terms of non-essential amino acids. In terms of essential amino acids, NEC treatment outperformed alcalase but not flavourzyme. The total free amino acid content of the hydrolysates was in the range of 53.01–90.37 mg/g, with hydrolysate alcalase hydrolysis being the lowest and NEC hydrolysis being the highest. Flavourzyme hydrolysis yielded a total free amino acid content of 79.64 mg/g.

The observed results are comparable with previous studies. For instance, Leni et al. [16] used seven commercial enzymes for MW hydrolysis and reported free amino acid levels ranging from 5.07 to 126.60 mg/g. The same study reported that the highest free amino acid used alcalase hydrolysis. Another study by Tang et al. [18] reported a slightly higher free amino acid content of MW when hydrolyzed by a combination of alcalase and flavourzyme (111.70 mg/g) compared to NEC-treated MW (90.37 mg/g) in this study.

In our study, NEC-treated MW hydrolysate had a greater total essential amino acid content (20.87 mg/g) than alcalase hydrolysate (9.48 mg/g) but a lower total essential amino acid content than flavourzyme hydrolysate (28.97 mg/g). This demonstrates that the enzyme from nuruk successfully hydrolyzed the native MW protein, enhancing bioavailability. Furthermore, MW hydrolysate can be recommended as a protein supplement for people with slow food digestion, such as the elderly [46].

### 3.3. Biological Activities of MW Hydrolysate

#### 3.3.1. Antioxidant Activity

Free radical-induced oxidative stress has been linked to several disorders, including diabetes, neurodegenerative diseases, and cardiovascular ailments. Natural food antioxidants could contribute to the inhibition of free radicals in our bodies and minimize oxidative stress [47]. Enzymatic hydrolysis is known to generate low molecular weight peptides and hydrophobic amino acids with antioxidative activities that are otherwise encoded within the parent protein sequence [12,35,48]. In this study, enzymatic hydrolysis also enhanced the antioxidant activity of the MW protein.

ABTS radical inhibition activities of all MW hydrolysates showed concentration-dependent increases (Figure 2a). NEC MW hydrolysate exhibited the higher inhibition of ABTS radicals compared to defatted MW with an IC_50_ value of 3.07 mg/mL. The highest ABTS scavenging potential was registered by alcalase hydrolysate (IC_50_ 2.73 mg/mL) (Appendix A). This could be attributed to its ability to generate the highest concentration of short-chain molecular weight peptides (≤1 kDa and 1–6.7 kDa) (Figure 2b and Appendix A). Overall, these observed findings support the idea that enzymatic hydrolysis improves the free radical scavenging ability of MW proteins [12,48].

#### 3.3.2. ACE Inhibitory Activity

Hypertension is one of the most common chronic medical conditions characterized by persistently elevated blood pressure. In humans, ACE plays a key role in blood pressure regulation, as well as in water and fluid balance [49]. Protein hydrolysate with ACE inhibition properties could be used as an alternative drug for hypertension with minimized side effects. Previous studies have suggested that the enzymatic hydrolysis of insect proteins could produce ACE-inhibitory hydrolysates [49,50].

In this study, we discovered the presence of ACE inhibitory action in the NEC hydrolysate of defatted MW for the first time. The ACE inhibitory activity of MW hydrolysate is summarized in Figure 2b. All enzyme hydrolysates exhibited ACE inhibitory activity. The ACE inhibition of hydrolysate decreased in the order of MW alcalase > Nuruk extract concentrate > MW flavourzyme hydrolysate. MW-A and MW-NEC were the most effective at inhibiting ACE, with IC_50_ values of 0.05 and 0.15 mg/mL, respectively (Figure 3b).

These results correlate with the molecular weight distribution data, in which MW-A and MW-NEC had a higher level of low molecular weight peptides (1–6.7 kDa) (Appendix A). Therefore, the ACE inhibition property was attributed to the peptides generated during enzymatic hydrolysis since the native protein in MW exhibited a very low ACE inhibition compared to the enzymatic hydrolysate. This property was closely related to the low molecular weight of the peptides produced through hydrolysis. On the other hand, the lower ACE inhibition from MW-F hydrolysate may be due to MW-F being rich in free amino acids rather than short-chain peptides, which are directly linked to ACE inhibition (Appendix A).

Previously, similar ACE inhibitory activities of insect protein hydrolysates were reported [2,11,49,50]. For instance, Sungwon et al. [2] indicated that the hydrolysis of MW with alcalase and flavourzyme improved ACE inhibitory activity with an IC_50_ of 0.047 mg/mL, which is in agreement with the present study. Hall et al. [50] also reported the ACE inhibition of cricket protein hydrolysate with IC_50_ ranging from 0.051–0.089 mg/mL. Overall, these results suggest that the NEC-treated MW protein could serve as an antihypertensive component in functional food and nutraceuticals. Moreover, this finding could pave the way to elucidate the active peptides responsible for ACE inhibition.

### 3.4. Sensory Properties of MW Hydrolysate

The sensory properties of hydrolysate are important when discriminating the bitter taste generated from enzymatic hydrolysis and evaluating an improvement in taste due to the tastes of active amino acids in the hydrolysate [51]. The e-tongue taste score result showed that all enzymatic hydrolysis improved in sweetness and saltiness while reducing the sourness of defatted MW. On the other hand, only nuruk extract hydrolysis improved the umaminess and contained a lower level of bitterness when compared with alcalase and flavourzyme hydrolysates. In contrast, hydrolysis with alcalase and flavourzyme experienced an increased bitterness.

The bitter taste in protein hydrolysate was mainly due to the formation of low molecular weight peptides composed primarily of hydrophobic amino acids by enzyme hydrolysis [51]. The data from the electronic tongue showed that alcalase and flavourzyme treatment increased bitterness whereas the nuruk extract concentrate did not. This was highly correlated with the proportion of low molecular weight peptides (≤1 kDa) in the molecular weight distribution data (Figure 2b). Alcalase hydrolysate scored a higher bitterness (8.8) in the e-tongue analysis (Figure 4), which could be related to its endopeptidase property that hydrolyzes the hydrophobic amino acid residues, leaving the nonpolar amino acid residues at the C-terminus of the peptide produced [35], whereas the bitterness of the flavourzyme hydrolysate was lower than that of alcalase. This could be attributed to the enzyme’s endo and exopeptidase nature, which selectively releases N-Terminal amino acid from polypeptides and proteins. The lower bitterness in NEC hydrolysate could be explained by nuruk, which is rich in different proteases such as acid protease and carboxypeptidase [35,52,53]. It is also rich in amino acid content (Table 5) that can decrease the bitterness of the hydrolysate [54]. This would improve the practical application of MW protein hydrolysates as a food ingredient.

## 4. Conclusions

To assess its potential for use in the food industry, this study tested the efficiency of using the nuruk extract concentrate as an enzyme to separate and hydrolyze proteins from *Tenebrio molitor* (mealworm larvae). It was demonstrated through enzymatic protein hydrolysis that the nuruk extract concentrate (10%) had approximately equivalent protease activity to that of alcalase (1%) and flavourzyme (1%), resulting in a comparable amount of hydrolysis and hydrolysis yield. Compared to commercial enzymes, it also generated more free amino acids, showing comparable bioactivities, including antioxidant and ACE inhibition, and improved taste profiles. Therefore, MW hydrolysate is incredibly beneficial as an alternative functional protein supplement. Moreover, these findings suggest that nuruk extract concentrate may be a promising alternative to commercial enzymes for mealworm hydrolysis. In general, this research represents the first investigation into the use of the nuruk extract concentrate for the enzymatic hydrolysis of mealworm protein, highlighting its potential as a valuable tool in the food industry. Future studies focusing on the identification of specific proteases in nuruk extract concentrate and exploring its potential application in hydrolyzing other dietary proteins are recommended.

## Figures and Tables

**Figure 1 foods-12-02188-f001:**
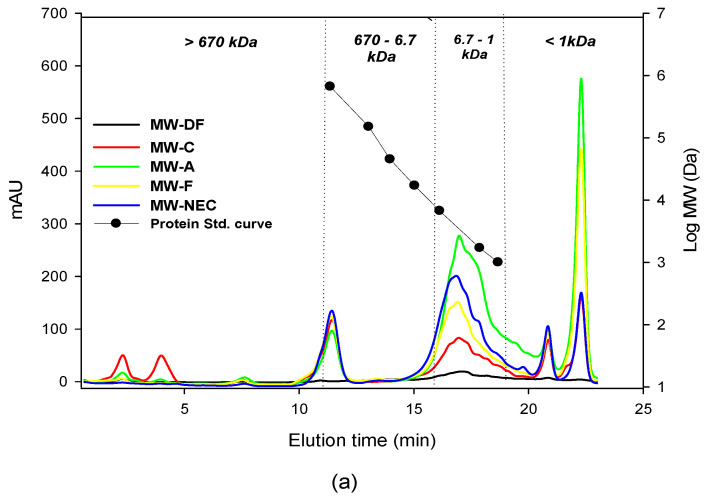
Size exclusion chromatographic analysis of mealworm hydrolysate. (**a**) SCE chromatogram of defatted mealworm, its hydrolysates, and standard mix. (**b**) Molecular weight (sum of peak area) for <1 kDa, 1–6.7 kDa, >670 kDa. MW-A: mealworm hydrolyzed by alcalase, MW-C: mealworm control (no enzyme), MW-DF: defatted mealworm, MW-F: mealworm hydrolyzed by flavourzyme MW-NEC: Mealworm hydrolyzed by nuruk extract concentrate.

**Figure 2 foods-12-02188-f002:**
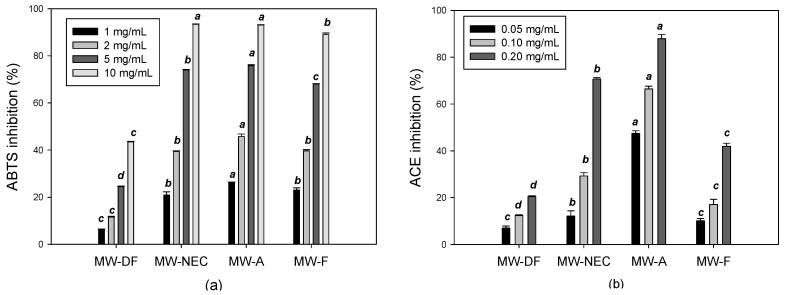
Biological activities of MW hydrolysate. ABTS radical inhibition of MW hydrolysate at different concentrations (**a**) and ACE inhibition of MW hydrolysate at different concentrations (**b**). MW-A: mealworm hydrolyzed by alcalase, MW-DF: Mealworm control, MW-F: mealworm hydrolyzed by flavourzyme, MW-NEC: mealworm hydrolyzed by nuruk extract concentrate. Different letters (a–d) on bars in a group show significantly different means.

**Figure 3 foods-12-02188-f003:**
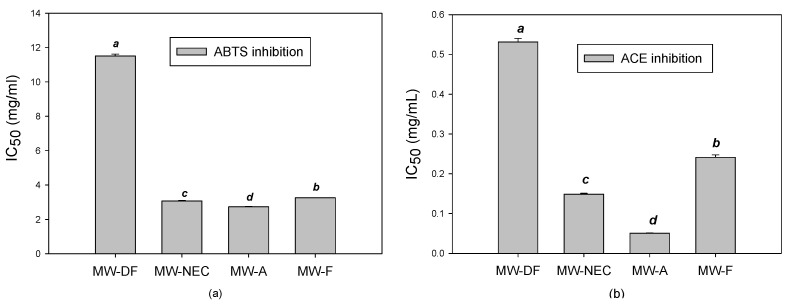
IC_50_ (mg/mL) values for the MW hydrolysate (**a**) IC_50_ for ABTS radical inhibition and (**b**) IC_50_ for ACE inhibition. MW-A: mealworm hydrolyzed by alcalase, MW-DF: Defatted mealworm, MW-F: mealworm hydrolyzed by flavourzyme, MW-NEC: mealworm hydrolyzed by nuruk extract concentrate. Different letters on bars in each figure show significantly different means.

**Figure 4 foods-12-02188-f004:**
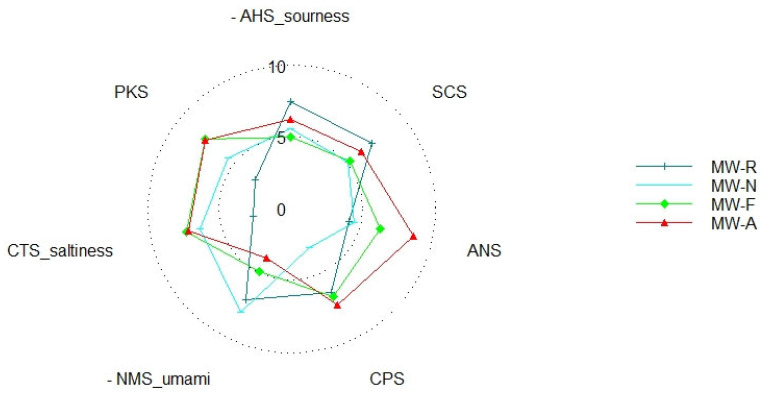
E-tongue taste score radar of defatted mealworm and its hydrolysate. MW-A: mealworm hydrolyzed by alcalase, MW-F: mealworm hydrolyzed by flavourzyme MW-R: Defatted mealworm, MW-N: mealworm hydrolyzed by nuruk extract concentrate. In the radar, PKS refers to sweetness, and ANS refers to bitterness.

**Table 1 foods-12-02188-t001:** Enzymatic hydrolysis treatments and their description.

Treatments	Description	E/S Ratio (*w*/*w*)
MW-C	Mealworm control	No enzyme
MW-NEC	Mealworm hydrolyzed by nuruk extract concentrate	10/100
MW-F	Mealworm hydrolyzed by flavourzyme	1/100
MW-A	Mealworm hydrolyzed by alcalase	1/100

**Table 2 foods-12-02188-t002:** Protease activity of crude nuruk extract and concentrate compared to commercial enzymes. Values in the table are the mean ± std. dev. (n = 3) and different letters within a column represent significantly different means (*p* ≤ 0.05).

Enzyme	Protease Activity (EU/mL)	Protease Activity (EU/g)
Nuruk crude extract	6.78 ± 0.07 ^c^	67.84 ± 0.73 ^d^
Nuruk extract concentrate (>30 kDa)	12.71 ± 0.33 ^a^	127.08 ± 3.30 ^c^
Alcalase 2.4 L (1%)	11.07 ± 0.07 ^b^	1107.27 ± 7.60 ^b^
Flavourzyme 1000 L (1%)	12.45 ± 0.02 ^a^	1245.37 ± 2.90 ^a^

EU/g refers to enzyme unit per gram, EU/mL refers to enzyme unit per mL.

**Table 3 foods-12-02188-t003:** Degree of hydrolysis, yield, and total soluble solids of enzymatic hydrolysis in mealworm samples.

Product	DH (%)	TSS (%)	Yield (%)
MW-C	0.51 ± 0.01 ^d^	3.60 ± 0.00 ^d^	18.62 ± 0.68 ^d^
MW-NEC	15.10 ± 0.08 ^a^	5.97 ± 0.06 ^b^	35.92 ± 0.78 ^b^
MW-A	9.56 ± 0.22 ^c^	7.97 ± 0.06 ^a^	42.88 ± 1.01 ^a^
MW-F	11.03 ± 0.46 ^b^	5.70 ± 0.10 ^c^	28.99 ± 0.71 ^c^

Different letters within a column represent significantly different means (*p* ≤ 0.05). MW-A: Mealworm hydrolyzed by alcalase, MW-C: Mealworm hydrolyzed without enzyme, MW-F: Mealworm hydrolyzed by flavourzyme, MW-NEC: Mealworm hydrolyzed by nuruk extract concentrate, TSS: Total soluble solid.

**Table 4 foods-12-02188-t004:** Proximate composition of whole and defatted MW (starting material for enzymatic hydrolysis) in %.

Parameter	MW-Whole	MW-Defatted
Crude protein	49.16 ± 0.17 ^b^	71.41 ± 0.09 ^a^
Crude fat	27.81 ± 0.73 ^a^	8.47 ± 0.02 ^b^
Moisture	3.41 ± 0.03 ^a^	0.29 ± 0.03 ^b^
Ash	3.09 ± 0.03 ^b^	4.63 ± 0.01 ^a^
Carbohydrate	16.52 ± 0.90 ^a^	15.19 ± 0.08 ^a^
Free amino acid	0.55 ± 0.01 ^b^	0.64 ± 0.03 ^a^
Total amino acid	39.62 ± 0.23 ^b^	53.48 ± 0.15 ^a^

Data with different letters represent significant difference at *p* ≤ 0.05. Nitrogen content from Kjeldahl analysis was multiplied by 6.25 to obtain protein content.

**Table 5 foods-12-02188-t005:** Free amino acid composition of raw and enzyme hydrolysate of defatted meal worm (mg/g).

Amino Acids	MW-DF	MW-NEC	MW-A	MW-F
His	0.47 ± 0.01 ^d^	3.42 ± 0.09 ^b^	1.86 ± 0.07 ^c^	3.35 ± 0.21 ^a^
Thr	0.05 ± 0.00 ^d^	1.12 ± 0.02 ^b^	0.49 ± 0.02 ^c^	2.28 ± 0.11 ^a^
Lys	0.19 ± 0.02 ^d^	3.35 ± 0.13 ^b^	1.36 ± 0.10 ^c^	5.49 ± 0.05 ^a^
Met	0.18 ± 0.00 ^b^	0.57 ± 0.01 ^a^	0.57 ± 0.02 ^a^	0.60 ± 0.05 ^a^
Val	0.39 ± 0.02 ^d^	4.55 ± 0.10 ^b^	2.56 ± 0.11 ^c^	5.58 ± 0.19 ^a^
Ilu	0.19 ± 0.01 ^d^	2.60 ± 0.07 ^b^	1.09 ± 0.06 ^c^	3.26 ± 0.10 ^a^
Leu	0.10 ± 0.01 ^d^	3.63 ± 0.10 ^b^	1.06 ± 0.05 ^c^	6.23 ± 0.18 ^a^
Phe	0.09 ± 0.00 ^d^	1.63 ± 0.04 ^b^	0.48 ± 0.02 ^c^	2.18 ± 0.13 ^a^
EAA	1.66 ± 0.08 ^d^	20.87 ± 0.54 ^b^	9.48 ±0.40 ^c^	28.97 ± 1.01 ^a^
Ser	0.57 ± 0.02 ^c^	1.09 ± 0.02 ^b^	0.33 ± 0.02 ^d^	2.15 ± 0.10 ^a^
Arg	0.26± 0.01 ^d^	6.37 ± 0.14 ^b^	3.30 ± 0.11 ^c^	7.76 ± 0.40 ^a^
Gly	0.08 ±0.00 ^d^	1.19 ± 0.03 ^b^	0.57 ± 0.03 ^c^	1.40 ± 0.07 ^a^
Asp	0.30 ± 0.00 ^d^	1.49 ± 0.04 ^a^	0.93 ± 0.03 ^b^	0.78 ± 0.03 ^c^
Glu	0.02 ± 0.01 ^d^	29.02 ± 1.20 ^a^	21.22 ± 1.62 ^b^	13.87 ± 0.52 ^c^
Ala	0.40 ± 0.03 ^c^	4.84 ± 0.13 ^a^	2.01 ± 0.09 ^b^	4.97 ± 0.12 ^a^
Pro	2.30 ± 0.11 ^d^	18.53 ± 0.47 ^a^	10.74 ± 0.57 ^d^	12.77 ± 0.41 ^b^
Cys	0.09 ± 0.01 ^d^	0.33 ± 0.02 ^c^	0.57 ± 0.02 ^a^	0.40 ± 0.01 ^b^
Tyr	0.75 ± 0.02 ^c^	6.62 ± 0.18 ^a^	3.86 ± 0.15 ^b^	6.57 ± 0.39 ^a^
NEAA	4.78 ± 0.21 ^d^	69.50 ± 1.01 ^a^	43.53 ± 0.57 ^c^	50.67 ± 2.01 ^b^
Total AA	6.44 ± 0.28 ^d^	90.37 ± 1.22 ^a^	53.01 ± 0.93 ^c^	79.64 ± 3.02 ^b^

EAA refers to essential amino acid, NEAA refers to non-essential amino acids, MW-A: mealworm hydrolyzed by alcalase, MW-DF: Defatted mealworm, MW-F: mealworm hydrolyzed by flavourzyme. MW-NEC: mealworm hydrolyzed by nuruk extract concentrate. Data with different letters represent significant difference at *p* ≤ 0.05.

## Data Availability

All the data are contained within the article or Appendix A.

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
