# Peer review of "Enzymatic Hydrolysis of Tenebrio molitor (Mealworm) Using Nuruk Extract Concentrate and an Evaluation of Its Nutritional, Functional, and Sensory Properties"

_foods, 2023, doi:10.3390/foods12112188_

Round 1
Reviewer 1 Report
Dear authors,
The production of cheap sources of protein is vital to support the constant growth of the human population and to ensure food security for future generations. The production of protein hydrolysates from Tenebrio molitor seems to be a very interesting and attractive topic to guarantee a cheap source of protein for human and animal food. In this sense, the manuscript submitted for consideration of the journal Foods "Enzymatic hydrolysis of Tenebrio molitor (Mealworm) using Nuruk extract concentrate and evaluation of its nutritional, functional, and sensory properties" would have been an excellent contribution due to the importance and novelty of the subject.
However, unfortunately, there are numerous inaccuracies in this manuscript.
Remarks:
1. The "Result" and "Discussion" sections are merged, which contravenes the Foods journal guidelines. But, most relevantly, the manuscript does not really present any discussion or debate of the results obtained. There is no comparison of the results obtained with other similar results, nor is there any attempt to explain the results obtained. For this reason, only 29 bibliographic citations are referenced, which is very poor for a journal that is in the first quartile.
2. Some methods are not sufficiently explained. For example, the use of the "E-tonge" does not explain the hedonic scale used (apparently it is a scale of 0-10, but it does not say what it refers to), nor the sensory descriptors used.
3. The formats of tables and figures are not those suggested by the template provided by Foods magazine.
4. To write the equations, it is necessary to use the Word equation editor,
5. You must thoroughly check the wording of the paper. Instead of writing large blocks of sentences, you should separate the paragraphs by topics and use 3-5 sentences per paragraph. This would increase the reading fluency and comprehension of the paper.
6. The rest of the minor points are highlighted directly in the attached document.
Therefore, it is suggested to amend all these points and resubmit the manuscript.
Regards,
The reviewer

It is suggested that the manuscript be thoroughly revised and rewritten.
Author Response
Reviewer's 1 Comments and response
- The "Result" and "Discussion" sections are merged, which contravenes the Foods journal guidelines. But, most relevantly, the manuscript does not really present any discussion or debate of the results obtained. There is no comparison of the results obtained with other similar results, nor is there any attempt to explain the results obtained. For this reason, only 29 bibliographic citations are referenced, which is very poor for a journal that is in the first quartile.
Response 1: Thank you for your detailed comments. We appreciate your feedback. Regarding the format, we understand your concern, and we would like to mention that we have found previously published papers in the food domain with the "results and discussion" combined. We opted for this format to enhance reader comprehension. We made efforts to enhance the discussion section by comparing our results with closely related studies and providing explanations for observed variations. For example, we compared the protease activity of nuruk in our study with that of related works and explained the variations observed. Similar revisions were made in Sections 3.1.2 (degree of hydrolysis), 3.1.3 (molecular weight distribution), 3.2.1 (protein content), 3.2.2 (amino acid composition), and 3.3.2 (ACE inhibition data). Additionally, we have increased the number of references cited in the paper from 30 to 54.
- Some methods are not sufficiently explained. For example, the use of the "E-tonge" does not explain the hedonic scale used (apparently it is a scale of 0-10, but it does not say what it refers to), nor the sensory descriptors used.
Response 2: We appreciated the comment. We included elaboration about the principle of electronic tongue and what scale refers to in the method section 2.6. ASTREE is an electronic tongue based on potentiometric measurement principle using taste sensing electrodes and dedicated to taste analysis.
The sensors are designed to mimic the human tongue, and they can differentiate between different tastes and flavors such as sweet, sour, salty, bitter, and umami based on the changes in electrical properties in response to different chemicals or molecules. Before starting analysis, conditioning, calibration, and diagnostic steps were performed using previously reported protocol. In summary, a solution of hydrochloric acid (0.01 mol/L) was used for conditioning and calibration, whereas hydrochloric acid (0.01 mol/L), sodium glutamate (0.01 mol/L), and sodium chloride (0.01 mol/L) were used for diagnostic step. The scale of all taste attributes has been defined between 0 and 10, where 0 refers to the least intense and 10 was the most intense taste perception. We also included reference for readers to have full understanding of the techniques.
- The formats of tables and figures are not those suggested by the template provided by Foods magazine.
Response 3: Yes it was. We corrected it accordingly.
- To write the equations, it is necessary to use the Word equation editor,
Response 4: It is right. The problem arose due to Microsoft Word compatibility while inserting equations. Now, it corrected as per your suggestion
- You must thoroughly check the wording of the paper. Instead of writing large blocks of sentences, you should separate the paragraphs by topics and use 3-5 sentences per paragraph. This would increase the reading fluency and comprehension of the paper.
Response 5: Thank you for your insightful comment and suggestion regarding the organization and structure of the paper. We have carefully reviewed the wording and organization of the paper and have made the necessary changes to improve the flow and readability. Specifically, we have separated the paragraphs based on topics and limited them to 3-5 sentences each, as you recommended. This adjustment has enhanced the reading fluency and overall comprehension of the paper. In addition, we will have English editing services from mdpi. We appreciate your valuable input in helping us improve the quality of our work.
Reviewer 2 Report
This paper mainly investigates the nutritional, functional, and sensory properties enzymatic hydrolysis of Tenebrio molitor (Mealworm), which is relatively interesting to readers, but not sufficiently discussed, and only describes the results, without elaborating the deeper mechanisms behind and it's future application.
Moderate editing of English language were needed
Author Response
1.This paper mainly investigates the nutritional, functional, and sensory properties enzymatic hydrolysis of Tenebrio molitor (Mealworm), which is relatively interesting to readers, but not sufficiently discussed, and only describes the results, without elaborating the deeper mechanisms behind and it's future application.
Response 1: Thank you for your comment. We appreciate the feedback and have taken similar suggestions from other reviewers into consideration. In response, we have made improvements to our discussion section by providing comparisons between our results and related research works. For example, we have compared the protease activity of nuruk in our study with that of related works and provided explanations for observed variations. We have also applied similar revisions to sections such as 3.1.2 (degree of hydrolysis), 3.1.3 (molecular weight distribution), 3.2.1 (protein content), 3.2.2 (amino acid composition), and 3.3.2 (ACE inhibition data). These revisions have helped enhance the quality and depth of our paper. Thank you for your valuable input.
- Comments on the Quality of English Language
Moderate editing of English language were needed
Response 2: Since we received similar comments from other reviewers, we decided to seek an English editing service from mdpi.
Reviewer 3 Report
This paper studied on hydrolysate of proteins from mealworm and characterization of their function. This is a common study that hydrolyzes proteins to exhibit some functionality. This study mentioned that Nuruk is better than other tested proteases. I think authors necessary to clarify the characteristics, not speculation.
Authors used mealworm in this study. What are the major proteins in mealworm?
Authors should mention the protein sources of functional peptides.
Table 2
Alcalase, 1% but U.mg does not match.
L140 and in many sentences, Authors should carefully write significant figures.
unify “hr” and “hour”
“10% of NEC would have equivalent activity with alcalase (1%) or flavorzyme (1%).”
The sentence mentioned the activity is the same, but Table showed a significant difference. I did not understand the relation.
E-tongue:
What is CPS and SCS?
What means the numbers in the figure 0, 5, and 10 represent?
molecular weight distribution data (Fig. 2 (b)).
Fig.2 b is correct?
What enzymes are contained in Nuruk?
Please clarify it is from a single fungus or multiple fungi. If multiple, it needed clarify rods or composition.
L468 “by nuruk being rich in different protease (both endo- and exopeptidase)” I think reference needed.
What is the relationship between brix and the sweetness of the E-tongue? I think data seem not match. Please explain the difference.
Some sentences were complicated and difficult to understand, but overall there is no problem. It is better to correct significant figures.
Author Response
Response to Reviewer 3 Comments
- Authors used mealworm in this study. What are the major proteins in mealworm? Authors should mention the protein sources of functional peptides.
Response 1: Thank you for your comments. We appreciate your insights. The proteins found in mealworms encompass a variety of types, including structural proteins, storage proteins, transport proteins, and enzymes. Among these, structural proteins like cuticle exhibit poor solubility in water, which can be improved through enzymatic hydrolysis. In our study, we focused on investigating various quality characteristics of mealworm hydrolysate as a food material. While the identification of functional peptides holds significance, it falls outside the scope of our current research and could be recommended for future studies. Our main objective was to demonstrate the excellent value of mealworm hydrolysate as a food material, rather than specifically fractionating peptides to assess their functions. We have explicitly mentioned these points in the introduction section of our paper. Thank you for your valuable feedback.
- Table 2: Alcalase, 1% but U.mg does not match.
Response 2. Yes, it is right and we corrected it on the table.
- L140 and in many sentences, Authors should carefully write significant figures.
Response 3: That is a valid comment. We corrected them and use similar significant figures in the article
- unify “hr” and “hour”
Response 4. Thank you for the comment. We have corrected the inconsistency throughout the document.
- “10% of NEC would have equivalent activity with alcalase (1%) or flavorzyme (1%).”
The sentence mentioned the activity is the same, but Table showed a significant difference. I did not understand the relation.
Response 5: Thank you for your comment. We acknowledge that there are statistically significant differences between the values obtained for protease activity. We have addressed this concern by rectifying the term "approximately equivalent," as it was initially used to estimate the nuruk enzyme dose for the experiment. The TNBS method was employed to determine the protease activity of each enzyme and subsequently used to estimate the approximate amount of enzyme required for mealworm hydrolysis. While the values measured in units per milliliter were found to be significantly different, we suggest that a rough approximation can be made, such as setting 10% NEC (Nuruk Enzyme Concentrate) approximately equal to 1% alcalase and 1% flavourzyme, based on their respective protease activities. Thank you for bringing attention to this point, and we have made appropriate clarifications in the revised version of our manuscript.
- E-tongue: What is CPS and SCS?. What means the numbers in the figure 0, 5, and 10 represent?
Response 6: Thank you for your question. CPS and SCS serve as auxiliary sensors for taste patterning. In Alpha-MOS electronic tongue instrument, these sensors generate data on taste values, which are then digitized and subjected to various statistical analyses, such as taste screening and principal component analysis (PCA). It is important to note that taste parameters such as salty, sour, and umami tastes share similar taste mechanisms across different samples, suggesting the involvement of a single sensor for these flavors. However, other tastes such as sweet, bitter, astringent, and fishy tastes have distinct taste mechanisms that vary with different samples. For instance, the bitterness of medicine differs from that of coffee, and the sweetness of sugar varies from that of fruit. In cases where a single sensor is inadequate to express the complexities of a particular sample, multiple sensors are utilized to capture the taste patterns accurately. We have included additional references in the revised manuscript to provide readers with further explanation and support. To avoid any confusion, we have decided to exclude CPS and SCS from the graphical representation. If deemed necessary, we are open to using the table below instead of a graph. In the figure, the numbers represent the intensity of taste perception, with 0 indicating the least intense and 10 denoting the highest intensity. Thank you for your feedback, and we will ensure that these clarifications are clearly conveyed in the revised version of our article.
Product |
AHS (Sourness) |
PKS (Sweetness) |
CTS (Saltiness) |
NMS (Umaminess) |
ANS (Bitterness) |
MW-R |
7.4 |
3.1 |
2.7 |
7.1 |
4.2 |
MW-NEC |
5.5 |
5.6 |
6.5 |
8 |
4.6 |
MW-F |
4.9 |
7.7 |
7.4 |
5 |
6.4 |
MW-A |
6.2 |
7.6 |
7.3 |
3.9 |
8.8 |
- Molecular weight distribution data (Fig. 2 (b)). Fig.2 b is correct?
Response 7: We sincerely appreciate the reviewer's feedback and the opportunity to address the confusion that may have arisen. We have made the necessary revision to the paragraph explaining Fig. 2b. To enhance understanding of how we obtained the value of the eluted peptide area in Figure 2b, we have included a division and labeling of the area in Figure 2a. Additionally, we have explained how the data was generated, specifically by summing the peak areas (not peak height) of < 1 kDa, 1 - 6.7 kDa, and > 670 kDa. These modifications aim to provide a clearer representation of the methodology and data analysis employed in our study. Thank you for bringing this to our attention, and we have taken your suggestion into careful consideration in the revised version of our manuscript.
- What enzymes are contained in Nuruk? Please clarify it is from a single fungus or multiple fungi. If multiple, it needed clarify rods or composition.
Response 8: We appreciate your comment and question. Nuruk represents a diverse microbial ecosystem that undergoes natural fermentation using various ingredients, resulting in a complex mix of strains. The microbial ecology of Nuruk is indeed a complex subject that requires specialized study. In our research, we chose to utilize Nuruk as a substitute for commercial enzymes to take advantage of the diverse and complex enzyme composition present in Nuruk, which could offer a broader range of hydrolysate products during the mealworm hydrolysis process.
However, since our primary focus was on utilizing the protein component of mealworms, we conducted a comparison of the enzyme activity between Nuruk and commercial enzymes using a standardized approach to demonstrate the presence of proteolytic activity (as described in Section 3.1.1). To provide a more comprehensive understanding, we have included additional details in the introduction section and have expanded upon the results of the protease activity section, specifically highlighting the characteristics and role of Nuruk in the process.
Thank you for your valuable input, and we have incorporated these clarifications into the revised version of our manuscript.
Top of Form
- L468 “by nuruk being rich in different protease (both endo- and exopeptidase)” I think reference needed.
Response 9: We appreciate your suggestion, as it has highlighted the need for additional references in our manuscript. In other research papers, carboxypeptidase and acidic protease activities have been measured to assess the enzymatic activity of nuruk. To address this, we have included a reference that explains how nuruk produces free amino acids and peptides from raw materials during the alcohol manufacturing process. Taking these research findings into account, we have modified the sentence to state that nuruk is presumed to possess both endo and exo enzymatic activities. These revisions have been made to provide a more comprehensive understanding of the enzymatic capabilities of nuruk. Thank you for bringing this to our attention, and we have made the necessary updates in the revised version of our manuscript.
- What is the relationship between brix and the sweetness of the E-tongue? I think data seem not match. Please explain the difference.
Response 10: Thank you for your comment. In response to your feedback, we have explained the measurement of total soluble solids in Section 3.1.2. Total soluble solids refer to the overall amount of solids that can dissolve in water. Typically, in fruits, sugars such as fructose, glucose, and sucrose are the predominant soluble solids, and they serve as indicators of sweetness and fruit quality. However, in our study, we employed total soluble solids as an index to measure the yield of solubilized protein components from mealworms. A study conducted by Jung (2020) utilized total soluble solids to measure yield during the enzymatic and high-pressure extraction of white-spotted radish.
Therefore, it is important to note that the sweetness observed in the results obtained from the electronic tongue (E-tongue) reflects the measurement of free sugars, which are the sweet components among the soluble solids. On the other hand, the total soluble solids we measured primarily reflect the content of solubilized amino acids. As a result, there is no correlation between the sweetness measured by the E-tongue and the total soluble solids reported in our study. We appreciate your attention to this matter, and we have included these clarifications in the revised version of our manuscript.
- Comments on the Quality of English Language
Some sentences were complicated and difficult to understand, but overall there is no problem. It is better to correct significant figures.
Response: Since we received similar comments from other reviewers, we are planning to request an English editing service and we corrected the significant figures too.
Round 2
Reviewer 1 Report
The authors accepted most of the points made and I believe that the work can now be published.
Author Response
Thank you very much!
Reviewer 2 Report
Generally,the manuscript is interesting to readers, has been carefully revised and improved a lot. several correction is still need to made before publication.
1. In table 2, line 299 protetease activity (EU/ mL enzym) is confusing, it should be misspelling
2. line 266, line 299, line 336 et al. As shown in the statistical analysis method, p ≤ 0.05 were considered significant. the express of p ≤ 0.05 should be same in the tables or figures within the whole paper
3. line 299, line 352, molecule weight unit using KDa or KD should be unifiled within the whole paper. please check
4. line 381, Figure 1 , line 511 figure 2, line 548 figure 3, are not clear to read. please amplify the size and improve the resolution ratio of the figure
5. line 409-410 , writing style of MW- Whole and MW defatted should be unifiled, such as MW - Defatted
6. The format of Table 5, line 489-490 , especially in the second column of MW-DF should be checked
7. Please explain more, what is the reason to choose different concentration of MW-DF, MW-NEC et al. when measuring the ABTS inhibition and ACE inhibition?
Minor editing of English language required
Author Response
This paper mainly investigates the nutritional, functional, and sensory properties enzymatic hydrolysis of Tenebrio molitor (Mealworm), which is relatively interesting to readers, but not sufficiently discussed, and only describes the results, without elaborating the deeper mechanisms behind and it's future application.
Response: We greatly appreciate the valuable feedback provided by the reviewers and acknowledge the importance of discussing the underlying mechanisms and future applications of our research. In response to these comments, we have made significant revisions to our manuscript, aiming to provide a more comprehensive analysis of our findings.
To address the concern regarding the underlying mechanisms, we have expanded the discussion section. We now provide a thorough analysis of the protease activity and properties of nuruk, based on previously published works (section 3.1.1, lines 257-259 and 270-275). However, due to nuruk being a complex microbial ecosystem involved in natural fermentation, we couldn't specify the precise mode of action. As mentioned in the conclusion (section 4, lines 521-523), future research will focus on studying specific enzymes and their mode of action. We apologize for any confusion caused by this limitation.
Furthermore, we have highlighted the potential of substituting commercial enzymes with nuruk by evaluating its hydrolytic properties for protein breakdown in MW. We have expanded the discussion on the degree of hydrolysis (lines 286-287 and 295-298) and conducted a comprehensive analysis of total soluble solids and molecular size (completely rewritten) to demonstrate the effectiveness of nuruk in protein breakdown. By providing this detailed information, we aim to offer readers a clearer understanding of the potential and optimal conditions for nuruk hydrolysis. Additionally, we have outlined the practical implications of using nuruk instead of commercial enzymes in the food industry, emphasizing the potential cost reduction and availability of an affordable product.
Lastly, we have evaluated the nutritional composition of mealworm hydrolysate, considering its free amino acid profile, biological activity, and sensory properties. This analysis provides insights into the nutritional and health benefits of mealworms as a protein source, and the sensory evaluation allows us to assess palatability and acceptability, which are important factors influencing future applications of mealworm hydrolysate. We have expanded on this discussion (lines 352-383) to demonstrate the potential of mealworms as a protein source based on protein and amino acid data, compared to dietary recommendations.
In conclusion, we have taken the reviewers' suggestions into account and made substantial revisions to our manuscript. The discussion section now delves deeper into the mechanisms underlying enzymatic hydrolysis of Tenebrio molitor and provides a comprehensive analysis of its future applications. We firmly believe that these additions have significantly enhanced the value and overall quality of the paper. We sincerely thank the reviewers for their constructive feedback, as it has undoubtedly improved the scientific rigor and relevance of our study.
Reviewer 3 Report
The revised MS was quite improved and reflected reviewer's suggestion.
Some grammatic error still remained.
Author Response
We corrected all grammatical and spelling errors.
Since there was no specific issue mentioned, we tried to improve result and discussion part, especially in the SEC section, lines 352 -366, 428 - 434, and 443 - 448 by re-writing to correct vague and incoherent sentences.